# In Case of Fire, Escape or Die: A Trait-Based Approach for Identifying Animal Species Threatened by Fire

Eugênia K. L. Batista [1,2,*], José E. C. Figueira [2], Ricardo R. C. Solar [3], Cristiano S. de Azevedo [4], Marina V. Beirão [5], Christian N. Berlinck [6], Reuber A. Brandão [7], Flávio S. de Castro [5], Henrique C. Costa [8], Lílian M. Costa [9], Rodrigo M. Feitosa [10], André V. L. Freitas [11], Guilherme H. S. Freitas [12], Conrado A. B. Galdino [13], José E. Santos Júnior [14], Felipe S. Leite [15], Leonardo Lopes [16], Sandra Ludwig [1], Maria C. do Nascimento [17], Daniel Negreiros [1], Yumi Oki [1], Henrique Paprocki [18], Lucas N. Perillo [1], Fernando A. Perini [17], Fernando M. Resende [1], Augusto H. B. Rosa [11], Luiz F. Salvador, Jr. [19], Larissa M. Silva [18], Luis F. Silveira [20], Og DeSouza [21], Emerson M. Vieira [22] and Geraldo Wilson Fernandes [1,23]

1   Laboratório de Ecologia Evolutiva & Biodiversidade, Departamento de Genética, Ecologia e Evolução/ICB, Universidade Federal de Minas Gerais, Belo Horizonte 31270-901, MG, Brazil
2   Laboratório de Ecologia de Populações, Departamento de Genética, Ecologia e Evolução/ICB, Universidade Federal de Minas Gerais, Avenida Presidente Antônio Carlos, 6627, Belo Horizonte 31270-901, MG, Brazil
3   Centro de Sínteses Ecológicas e Conservação/ICB, Universidade Federal de Minas Gerais, Avenida Presidente Antônio Carlos, 6627, Belo Horizonte 31270-901, MG, Brazil
4   Laboratório de Zoologia dos Vertebrados, Departamento de Biodiversidade, Evolução e Meio Ambiente/ICEB, Universidade Federal de Ouro Preto, Campus Morro do Cruzeiro, s/n, Bauxita, Ouro Preto 35400-000, MG, Brazil
5   Laboratório de Ecologia de Insetos, Departamento de Genética, Ecologia e Evolução/ICB, Universidade Federal de Minas Gerais, Avenida Presidente Antônio Carlos, 6627, Belo Horizonte 31270-901, MG, Brazil
6   Centro Nacional de Pesquisa e Conservação de Mamíferos Carnívoros, Instituto Chico Mendes de Conservação da Biodiversidade, Estrada Municipal Hisaichi Takebayashi, 8600, Bairro da Usina, Atibaia 12952-011, SP, Brazil
7   Laboratório de Fauna e Unidades de Conservação, Departamento de Engenharia Florestal, Universidade de Brasília, Brasília 70910-900, DF, Brazil
8   Departamento de Zoologia, Universidade Federal de Juiz de Fora, Rua José Lourenço Kelmer, s/n, São Pedro, Juiz de Fora 36036-900, MG, Brazil
9   Espinhacensis Pesquisas Ambientais, Brumadinho 35460-000, MG, Brazil
10  Laboratório de Sistemática e Biologia de Formigas, Departamento de Zoologia, Universidade Federal do Paraná, Av. Cel. Francisco Heráclito dos Santos, C.P. 19020, Curitiba 81531-980, PR, Brazil
11  Departamento de Biologia Animal e Museu da Diversidade Biológica, Instituto de Biologia, Unicamp, Campinas 13083-862, SP, Brazil
12  Departamento de Ecologia, Instituto de Ciências Biológicas, Universidade Federal de Goiás, Avenida Esperança s/n, Campus Samambaia, Goiânia 74690-900, GO, Brazil
13  Programa de Pós-graduação em Biologia de Vertebrados, Pontifícia Universidade Católica de Minas Gerais, Avenida Dom José Gaspar, 290, Belo Horizonte 30535-901, MG, Brazil
14  Laboratório de Biodiversidade e Evolução Molecular, Departamento de Genética, Ecologia e Evolução/ICB, Universidade Federal de Minas Gerais, Avenida Presidente Antônio Carlos, 6627, Belo Horizonte 31270-901, MG, Brazil
15  Sagarana Lab. IBF, Universidade Federal de Viçosa, Campus Florestal, Rodovia LMG-818, km 6, Florestal, 35690-000, MG, Brazil
16  Laboratório de Biologia Animal, IBF, Universidade Federal de Viçosa, Campus Florestal, Rodovia LMG-818, km 6, Florestal, 35690-000, MG, Brazil
17  Laboratório de Evolução de Mamíferos, Departamento de Zoologia/ICB, Universidade Federal de Minas Gerais, Avenida Presidente Antônio Carlos, 6627, Belo Horizonte 31270-901, MG, Brazil
18  Museu de Ciências Naturais, PUC Minas, R. Dom José Gaspar, 290, Belo Horizonte 30535-901, MG, Brazil
19  Neotropical Research—Biodiversidade e Conservação, Rua Henrique Passini 290/302, Belo Horizonte 30220-380, MG, Brazil
20  Seção de Aves, Museu de Zoologia da Universidade de São Paulo, Avenida Nazaré 481, Ipiranga 04263-000, SP, Brazil
21  Laboratório de Termitologia, Departamento de Entomologia, Universidade Federal de Viçosa, Viçosa 36570-900, MG, Brazil
22  Departamento de Ecologia, Universidade de Brasília, Brasília 70910-900, DF, Brazil
23  Knowledge Center for Biodiversity, 31270-901 Belo Horizonte, MG, Brazil
*   Correspondence: biogenia.k@gmail.com

**Abstract:** Recent studies have argued that changes in fire regimes in the 21st century are posing a major threat to global biodiversity. In this scenario, incorporating species' physiological, ecological, and evolutionary traits with their local fire exposure might facilitate accurate identification of species most at risk from fire. Here, we developed a framework for identifying the animal species most vulnerable to extinction from fire-induced stress in the Brazilian savanna. The proposed framework addresses vulnerability from two components: (1) exposure, which refers to the frequency, extent, and magnitude to which a system or species experiences fire, and (2) sensitivity, which reflects how much species are affected by fire. Sensitivity is based on biological, physiological, and behavioral traits that can influence animals' mortality "during" and "after" fire. We generated a Fire Vulnerability Index (FVI) that can be used to group species into four categories, ranging from extremely vulnerable (highly sensible species in highly exposed areas), to least vulnerable (low-sensitivity species in less exposed areas). We highlight the urgent need to broaden fire vulnerability assessment methods and introduce a new approach considering biological traits that contribute significantly to a species' sensitivity alongside regional/local fire exposure.

**Keywords:** fire ecology; resilience; sensitivity; functional traits; savanna ecosystems; species vulnerability; fauna; fire exposure; index

## 1. Introduction

Natural fire has shaped species evolution in savanna ecosystems worldwide [1,2]. In these ecosystems, animal species are relatively tolerant to low-severity and patchy fires. Natural fires usually allow individuals to survive or reestablish populations from adjacent preserved ecosystems after burning. However, extreme wildfires have resulted from a synergy between severe droughts, high temperatures, low air humidity, windy days, and increased human ignitions, which increase the flammability of terrestrial ecosystems and expand fire's niche around the world [3–6]. Recent studies show alarming prospects, suggesting that under different future climate scenarios, changes in fire regimes are expected in the 21st century in terms of a meaningful increase, variability, and frequency of extreme events posing a major threat to global biodiversity [6,7].

Fire affects animal populations directly and indirectly [8–12]. During the fire event, animals that are unable to flee or seek appropriate shelter may die due to smoke inhalation, radiant heat, or by being directly killed by flames [11,13,14]. The fire-induced stress is likely mediated by the species' biological traits (e.g., ability to flee or use non-flammable refugia) [10], environment properties (e.g., refuge availability) [15–17], and fire behavior (e.g., fire intensity and severity). After burning, fire-induced changes to the landscape often result in indirect effects related to microclimatic deterioration and loss of food, habitat, and shelter. Populations of some species may decline immediately following the fire, probably due to the increased direct mortality and emigration [13], whereas others can persist in the burned areas for longer periods, be favored by wildfires [18], or even grow in abundance if the fire event happens to eliminate their stronger competitors [19].

Incorporating species' physiological, ecological, and evolutionary characteristics with their local fire exposure might facilitate more accurate identification of the species most at risk from fire. In this paper, we propose a framework for identifying the animal species most vulnerable to local extinction from fire-induced stress in the Brazilian savannas. This framework might be useful to guide users to independently measure two dimensions of vulnerability to fire, namely sensitivity (the lack of potential for a species to persist in situ and its inability to avoid the direct or indirect effects of fire) and exposure (the extent to which each species' physical environment is altered). These two dimensions can then be used to rank species according to their fire vulnerability.

## 2. Methods

In this study, vulnerability results from the interaction between two main components: exposure and sensitivity [20,21]. *Exposure* refers to the presence, extent, and magnitude to which a system or species experiences stressors [20]. *Sensitivity* is the degree to which they are affected or harmed by exposure to stressors. The sensitivity of a species or an individual is determined by intrinsic factors (e.g., physiological tolerance or behavior), which are inherently mediated by resistance and adaptive capacity or resilience.

### 2.1. Fire Exposure

In Brazilian savannas, lightning may cause natural fires when there is little or no precipitation immediately following ignition [22]. In these open landscapes, natural fires are usually of low intensity and spread through the surface, burning litter, grasses, shrubs, and lower tree branches. This type of fire tends to be naturally extinguished at forest edges, where there is high moisture and low fine fuel at the ground level [23]. However, even if the fire advances into a relatively undamaged forest patch, the short-term effects are likely to be minimal because flames spread with low intensity in lines about 10 cm high, burning just above the leaf litter in the soil [24]. Low-intensity fires usually do not drastically increase the temperature below ground and do not reach or ignite the canopy of taller trees or peaty soils. It spreads across the surface irregularly while leaving patches of vegetation unburned, which promotes small-scale heterogeneity and provides shelter and distinct microclimatic conditions that enable individuals to survive in the post-fire landscape [17,25–27].

However, both the increasing number of human ignitions and man-made changes in landscape flammability have altered fire behavior and seasonality, leading to the replacement of natural burns by catastrophic wildfires [28]. Because they occur late in the dry season, when the highest temperatures and lowest precipitation and fuel moisture are recorded, human ignitions find ideal conditions to spread, being able to quickly increase in intensity and extension [29]. With temperatures exceeding 600 °C, flames can severely burn permanent fire refugia (e.g., forest patches), reaching treetops and killing animals that seek security in the upper strata of vegetation, either from the extreme heat or from inhaling excess smoke [11,30]. Others that try to escape may become disoriented and get caught in the fire and even fossorial animals may die from excessive heat transfer through the soil [31]. In addition, wildfires usually burn homogeneously without leaving micro-refuges that would provide the fauna with habitat, shelter from predators, and resources for feeding or nesting in the post-fire landscape [15].

In this framework, we address exposure by combining three fire regime parameters: (1) fire return interval, (2) fire extent, and (3) fire seasonality (as a proxy for fire severity). Fire return interval is a key parameter of many ecological processes [32] and refers to the recovery period available to plants and animals between consecutive fires. For the Brazilian savannas, it has been estimated that without human ignitions, the mean interval between successive fires would be 3–6 years [33]. To address this fire regime parameter, it will be necessary to generate annual fire maps and calculate fire return intervals across the entire study area. From this map, landscapes can be evaluated according to the percentage of fire-prone vegetation that is outside the expected thresholds for natural fire regimes (3–6 years). We recommend a minimum 20-year period for analysis. Fire extent is also an important parameter and is addressed here as the number of times the study area burned more than 10% of its total extent during the dry season over a 10-year period. We assume that fires during and around the rainy season tend to be of low severity, remaining within the tolerance thresholds of Brazilian savanna ecosystems. Fire seasonality is often delineated as a period of the year during which fires usually occur. Brazilian savannas are typically seasonal ecosystems, with two discrete periods of rain (November–June) and dryness (July–October). We propose that fire events should be divided according to the seasons during which they occur, assuming that weather conditions will often be associated with burn severity. Fire seasonality can be measured by summing the area burned (by

season) in all years (10-year period) and calculating the percentage burned in the dry season.

After characterizing the local fire regime, an area receives a sub-score for each parameter, as shown in Table 1, that is used to calculate the exposure factor.

**Table 1.** Fire exposure can be measured using three fire regime parameters and their respective scores to calculate the Fire Vulnerability Index.

| Fire Regime Parameter | Effect on Vulnerability/Score | | | | |
| --- | --- | --- | --- | --- | --- |
| | Negligible | Slight | Moderate | Important | Extreme |
| | **0.5** | **1.0** | **2.0** | **4.0** | **8.0** |
| Fire return interval [1] | <10% | 10–30% | 31–50% | 51–70% | >70% |
| Fire extent [2] | 0 | 1 | 2 | 3 | ≥4 |
| Fire seasonality [3] | <10% | 10–30% | 31–50% | 51–70% | >70% |

[1] Percentage of fire-prone vegetation that is outside the expected thresholds for natural fire regimes. For the Brazilian savannas, it has been estimated that without human ignitions, the mean interval between successive fires would be 3–6 years. Recommended period of analysis: 20 years or more. [2] Number of years the study area burned more than 10% during the dry season. We assume that fires during and around the rainy season tend to be of low severity, remaining within the tolerance thresholds of Brazilian savanna ecosystems. Recommended period of analysis: 10 years. [3] Percentage of the total area burned during the dry season over 10 years.

## 2.2. Fire Sensitivity

Through two workshops held in person and virtually in 2018 and 2020 and various other consultations, we conducted expert-based information gathering and, together with an extensive literature survey, we compiled a range of biological, physiological, and behavioral traits that can influence animals' mortality "during" and "after" fire. The workshops assembled a team of 39 experts whose collective experience encompassed the biology of a broad range of taxonomic groups and the ecology of fire in savanna ecosystems. These experts were asked to consider that fires can (i) cause injury to and death of plant tissues through heat, and animals through smoke inhalation, desiccation, contact with flames or thermal radiation, and (ii) change habitat structure or resource availability, promoting recovery delay, increased starvation, and predation, since it exposes nests, reduces nesting places and refuges both during and after the fire. A survey of the literature was performed in the Web of Science database (available at www.isiknowledge.com, accessed on 15 June 2023), considering publications from January 2010 to January 2023. We filtered the research based on keywords that were selected to identify studies on fire effects ('fire', 'wildfire') in multiple taxonomic groups ('beetle', 'bee', 'butterfly', 'termite', 'ant', 'bird', 'reptile', 'amphibian', 'flying mammal', 'small mammal', 'large mammal').

During a fire event, animals may be directly killed by anoxia, flame injury, extreme heat, or smoke poisoning. There are no animals that are completely resistant to fire, but some animals may have behavioral and morphological traits that confer them with better chances of survival, at least to low-intensity fire. After the fire event, animals may be indirectly affected by the simplified habitat structure with greater extremes of microclimate (e.g., temperature, humidity), restricted food resources (e.g., less abundant food, lower nutritional status and/or palatability), or interactions with other organisms (e.g., competition, increased predation, higher rates of parasitism). The traits are subjected to a zero-one score system depending on how that influences the species' sensitivity to fire: "one" for traits that increase the species' sensitivity (increased sensitivity) and "zero" for traits that make the species somewhat resistant or resilient to fire (decreased sensitivity). The species' sensitivity sub-score integrates the vulnerability index whose calculation will be described in detail below.

## 3. Results

Exposure should vary according to the study area, while sensitivity is inherent to the species and independent of where it lives. For that reason, only species sensitivity and the proposed vulnerability index will be addressed in the results section.

### 3.1. Sensitivity

Here, we compiled 14 species traits that can increase the vulnerability of the species to the direct and indirect effects of fire. These traits are explained in detail below and are summarized in Table 2.

**Table 2.** Animal functional traits associated with fire sensitivity or fire resistance/resilience.

| | Trait Group | Increase Vulnerability | Decrease Vulnerability |
|---|---|---|---|
| **DURING FIRE** | Dormancy | Species that often express deep torpor on flammable surfaces in the flame zone. | Non-hibernators; species that rapidly arouse from shallow torpor when exposed to smoke or flame noises; species that remain in torpor in places protected from fire, such as in deeper soil layers |
| | Escape decision | Animals that run away randomly when frightened; fossorial species with shallow burrowing behavior; species that take shelter in flammable or suffocating places, such as plants in the lower layers, litter, or cavities in small trees. | Animals that run toward nearby refuges when frightened; fossorial species with deep burrowing behavior; scansorial animals that seek refuge on top of tall trees during surface fires, in water, in termite mounds, or on rocky surfaces with little flammable material. |
| | Habitat use | Leaf litter-dwelling fauna in the o-horizon and other species that live or build nests in the lower strata of vegetation on flammable substrates, such as shrubs, grasses, dry and/or fallen trunks and branches, and small trees. | Soil-dwelling species that can burrow deeper into the ground; species that live or build nests close to perennial wetlands or water sources (semi-aquatic habits), below-ground, on rocky substrates, termite mounds with low flammability, and deep cavities inside massive tree trunks or in the upper strata of vegetation (on the top of tall trees). |
| | Mobility | Limited movement capability: slow-moving animals, weak flyers, ground-dwelling species that fail to climb trees, smaller jumpers with reduced effective jump height. | Good or excellent movement capability: fast runners, strong flyers, skilled climbers, larger jumpers with great effective jump height, and other jumping specialists that use catapult mechanisms. |
| | Morphology | Medium-bodied animals that may have difficulty fleeing or finding refuge; species whose bodies are covered with long, coarse fur or feathers. | Small-bodied animals that can find refuge more easily during a fire, while larger ones can flee or move away from affected areas; species with short fur, smooth skin, or covered with scales. |
| | Nest substrate | Species using flammable materials to build nests: thatched mounds, moss and lichen, fine grass or mammalian hair, and plant material such as bark, fiber, leaves, twigs, grasses, tussocks, and branches. | Species that use thermally insulating building materials: great amounts of soil in hard, protective clay mounds; species with deliberate behavior for modifying their surrounding environment causally reducing flammability; species that build subterranean nests without thatched mounds. |
| | Reproductive cycles | Synchronous reproduction, usually at the end of the dry season, exposing fragile life stages, pregnant, lactating, nesting, and brooding females to high-intensity fire. | Year-round breeders or species that reproduce during the wet season but decrease reproduction during the dry season. |
| | Sensory detection of fire cues | Species that spend most of their time in complex vegetation and rely primarily on the visual detection of fire (small-bodied animals could be even more vulnerable, as they usually have lower visual acuity). | Species that are able to detect olfactory and/or acoustic fire cues; species that can detect fire cues at lower thresholds; species that have thermoreceptors that can detect infrared radiation from fires; species relying primarily on the visual detection of fire, but that spend most of their time in the top of tall trees or open, low-stature vegetation and topographically simple landscapes. |
| | Social organization | Solitary animals or those that live in small family groups (parents and young); species with poorly developed social relationships (e.g., groups with weak connections) and whose individuals or groups lack effective communication skills. | Gregarious animals living in large groups; social species or those residing in more connected, reciprocal, and socially homogeneous groups. |
| | Behavioral plasticity | Late-successional species that require more structured habitats for nest sites and foraging, which take several years to recover. | Generalists that can temporarily adapt their diet and/or habitat preferences to the conditions and food resources available across the post-fire landscape; species that may benefit from fire-induced changes include early or mid-successional species. |

**Table 2.** *Cont.*

| | Trait Group | Increase Vulnerability | Decrease Vulnerability |
|---|---|---|---|
| **POST-FIRE** | Dormancy | Species that express multi-day torpor but need to rewarm frequently; species that use daily torpor, which is not as deep as hibernation, lasts only some hours rather than days or weeks, and is usually—but not always—interrupted by daily foraging and feeding. | Invertebrates that remain inactive after a fire, allowing their tissues to become desiccated (anhydrobiosis); invertebrates that express aestivation and remain in an inactive stage remarkably resistant to water loss; species that use multi-day torpor for weeks or even months after a fire or during fire season without the need to rewarm. |
| | Endogenous circadian rhythms | Diurnal ectotherms that depend on thermoregulation opportunities afforded by habitat structure; strictly diurnal prey species. | Nocturnal or crepuscular species; cathemeral or diurnal prey that can adjust their daily activity patterns. |
| | Mobility | Species with restricted home range; species with high site fidelity or territorial species; Migratory species (highly mobile), but with strong site fidelity. | Highly mobile species that travel long distances or show metapopulation dynamics; species with low site fidelity or non-territorial species. |
| | Morphology | Large ectotherms; invertebrates with thinner cuticles. | Large mammals; species capable of camouflaging in the scorched substrate; invertebrates with higher cuticle thickness. |

### 3.1.1. During Fire

High environmental temperatures predispose animals to heat stress, including physiological and behavioral disturbances such as hyperventilation and loss of coordination. For any species, there is a body temperature threshold beyond which cells undergo denaturation of proteins and membrane structures degrade, causing the individual's death [14,34]. Beyond the heat from flames, reduced oxygen and exposure to toxic compounds following smoke inhalation may be critical factors that increase animal mortality during a fire [35]. The longer an animal is exposed to high temperatures, anoxia, or smoke inhalation, the greater the chances of mortality; therefore, detecting and avoiding fire are essential behaviors for survival, especially for less mobile animals [31]. It is also important to mention the increased vulnerability of prey to opportunistic predators while trying to escape fires. The physical and behavioral traits that can increase or decrease a species' sensitivity to the direct effects of a fire are described in detail below and illustrated in Figure 1.

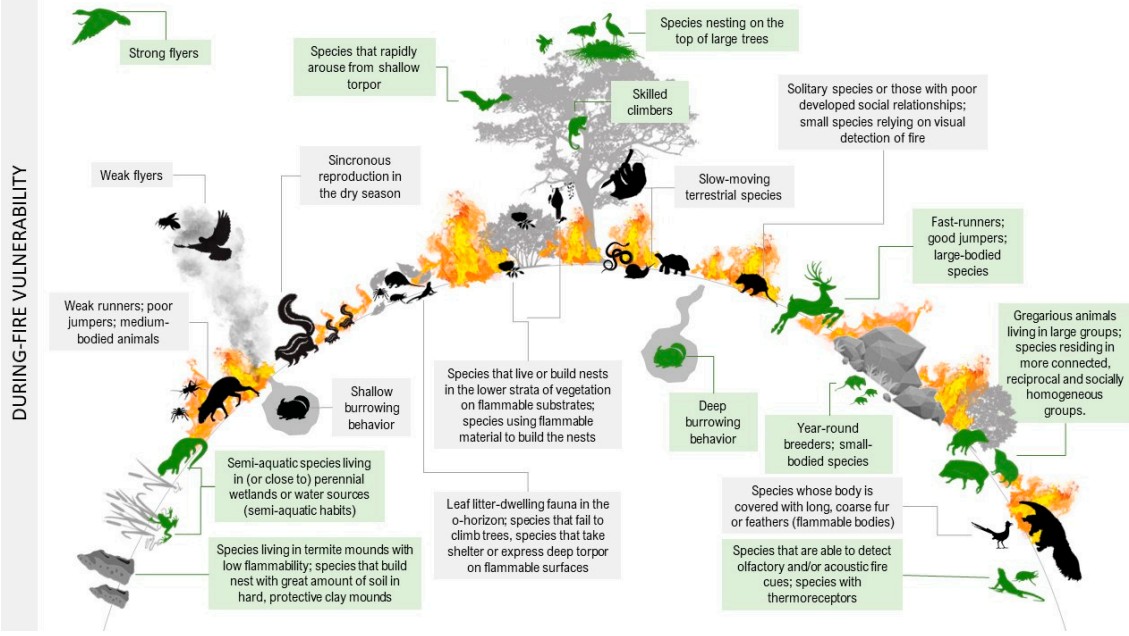

**Figure 1.** Fire vulnerability traits of species during wildfires. In green are animals that are most likely to survive the burning (decreased sensitivity). In black are animals whose traits increase their probability of death (increased sensitivity).

Dormancy

Being in a torpor during a wildfire may increase or decrease the chances of survival, depending on shelter security and depth of torpor. Low body temperatures are associated with decreased responsiveness (both sensory and locomotor function remains limited) and torpid animals might therefore face an increased mortality risk during fires due to inhalation of toxic smoke, oxygen depletion, and heat exposure [36]. Even though torpid animals can respond to fire stimuli, they may be slow in doing so; therefore, when in deep torpor animals are at risk of not responding to fire cues quickly enough to survive [37].

*Increased sensitivity:* Species that often present deep torpor on flammable surfaces in the flame zone [38].

*Decreased sensitivity:* Non-hibernators; species that rapidly arouse from shallow torpor when exposed to fire cues [37,39]; species that remain in torpor in places protected from fire, such as in deeper soil layers.

Escape Decision

When an animal becomes aware of approaching fire, it has two possible escape options: it can move away or find shelter. In the first approach, complex physiological adjustments, which include increases in oxygen consumption, body temperature, heart rate, and blood flow to skeletal muscle prepare the animal for prolonged strenuous activity [31]. As a result, the animal tends to move away from the threat. The alternative behavioral response involves stopping moving, bradycardia, and depression of metabolism [40]. As a result, the animal tends to hide in nearby refuges. Each decision will have different implications for the individual's survival.

*Increased sensitivity:* Species that run randomly when frightened can become disoriented when surrounded by fire [35]. In this case, animal survival depends on speed, agility, spatial memory, and a good navigation capacity, which can be compromised when in panic (or under severe stress). However, not only extreme heat but also smoke can affect the animal's ability to navigate, causing disorientation while trying to escape [30,41,42]. Fossorial species with shallow burrowing behavior may also die, as fire can induce advective flows in soils (e.g., shallow-nesting mining bees) [43–46]. Species that take shelter in flammable or suffocating places, such as plants in the lower layers, litter, or cavities in small trees [46].

*Decreased sensitivity:* Species that run toward nearby refuges when frightened. In this case, animal survival depends on how protected refuges are. Small animals using deep burrows or termite mounds as main shelters (e.g., lizards that flee to termite mounds and soil burrows) [47]. Scansorial species that seek refuge on top of tall trees during surface fires, in water, or on rocky surfaces with little flammable material.

Habitat Use

Typically, savanna fires produce flames of 1–2 m in height, which consume all the herbaceous and most of the woody vegetation of about this height [48,49]. Depending on how the species uses the habitat for nesting, foraging, or shelter, it may be more or less exposed to fire. More vulnerable and less mobile stages (e.g., eggs, offspring, larvae, pupae, and pre-emergent adults found in nests) are particularly susceptible to burning. Nest residents may die from lethal substrate heating unless the nests are adequately insulated.

*Increased sensitivity:* Leaf litter-dwelling fauna in the O-horizon [50,51] and other species that live or build nests in the lower strata of vegetation on flammable substrates, such as shrubs, grasses, dry and/or fallen trunks and branches, and small trees. For example, macroinvertebrate detritivores such as millipedes (Diplopoda), woodlice (Isopoda), and fly larvae (Diptera: Nematocera); fungivorous such as fungus gnats (Diptera: Sciaridae), and predators such as spiders (Araneae), centipedes (Chilopoda), and ground beetles (Coleoptera: Carabidae) [51]; leaf-litter herpetofauna [52,53]. Above-ground nesters that use pre-existing cavities made by other organisms in the thinner branches and smaller trees [54]; species nesting in soil deposits in cracks and crevices of soil-limited landscapes that occur in rock outcrops [55].

*Decreased sensitivity:* Soil-dwelling species, such as earthworms and worm lizards, that are able to burrow deeper into the ground (10–20 cm deep) [56,57]; species that excavate, use pre-existing holes or natural cavities underground to take shelter, or build nests (below-ground nesters) [58]; species that live or build nests close to perennial wetlands or water sources (aquatic or semi-aquatic habits), on rocky substrates, termite mounds with low flammability, and deep cavities inside massive tree trunks or in the upper strata of vegetation (on the top of tall trees) [25,59–62].

Mobility

This trait is associated with the species' ability to flee from the flame zone. Species that exhibit more powerful and flexible movement capabilities should be better able to escape fires.

*Increased sensitivity:* Nonvolant species of relatively low vagility, including amphibians, snakes, small lizards with short limbs (slow-running animals), and slow-moving animals such as sloths, turtles, and molluscs in general [13,57,63–65]; larger arboreal animals that are expected to attach less well to surfaces and have more difficulty distributing loads uniformly across large contact areas [66]; ground-dwelling species that fail to climb trees [7,11]; smaller jumpers with reduced effective jump height [67,68]; weak flyers with shorter wings and smaller flight muscles that usually can only fly a short distance at lower strata of vegetation on the flame zone [69,70].

*Decreased sensitivity:* Fast runners that can reach higher maximum speeds and escape the flames or travel greater distances, increasing the chances of finding safe shelter away from the fire [71]; birds that can fly higher and avoid the rising column of gasses, smoke, ash, particulates, and other debris produced by a fire [72,73]; lizards with longer limbs, larger toe pads, and more lamellae can run faster, exert stronger cling forces, and perch higher [74]; skilled climbers able to reach the tops of taller trees (at least 4–5 m above) [7,34]; larger jumpers with great effective jump height and other jumping specialists (e.g., arthropods that use catapult mechanisms [67,68].

Morphology

Body size can influence the animal's ability to find shelter or flee during a fire.

*Increased* sensitivity: Medium-bodied species may have difficulty fleeing or finding refuge and are more susceptible to direct mortality during the fire or increased chances of predation following a fire [75]. Terrestrial mammals whose bodies are covered with long, coarse fur may be more affected by fire due to the greater flammability of their bodies [76].

*Decreased sensitivity*: Small-bodied species can find and move into safe micro-refugia (e.g., frogs) [77]; large-bodied species are able to flee or move readily away from the affected areas to avoid direct mortality [75].

Nest Substrate

Beyond location, the substrate used to build nests can increase or decrease the chances of fire spreading.

*Increased sensitivity:* Species that build nests with thatched mounds [78]; moss and lichen (dry out rapidly because they lack developed root systems) [55,79]; fine grass or mammalian hair (capable of trapping a great deal of air) [80]; plant material such as bark, vegetal fiber, leaves, twigs, grasses, tussocks, and branches [81,82].

*Decreased sensitivity*: Species that use a great amount of soil in the nests [83,84] to build hard, protective clay mounds (e.g., some termites, wasps, ants, and birds) [85]; species with a deliberate behavior for modifying their surrounding environment, thus reducing flammability (e.g., birds that reduce litter around their nests—'fuel management') [86,87]; species that build subterranean nests without thatched mounds [78].

Reproductive Cycles

Pregnant females may experience reduced speed, maneuverability, and endurance [88,89]. This likely occurs due to the additional physical load of the eggs or embryos, which makes the body broader and heavier [90–92] and decreases muscle strength [93]. Because they are slower and heavier, pregnant females tend not to endure long runs [94], which increases the likelihood of dying from fire injuries before finding shelter. In addition, pregnant and lactating females tend to spend most of their time stationary and sleeping, avoiding energetically costly behaviors such as running or climbing [95–97]. The longer vulnerable life stages are exposed to fire, the greater the risk of individual mortality and population decline.

*Increased sensitivity:* Species with synchronous reproduction at the end of the dry season, exposing pregnant, lactating, nesting, and brooding females to high-intensity fire (e.g., holometabolous insects whose larvae are restricted to dry-season) [98]. In species with a higher allocation of parental care, females may be burned in an attempt to protect the offspring or by delaying their decision to flee the fire [99]. K-strategists would be particularly disadvantaged because, in addition to longer pregnancies, parental care is more pronounced and offspring tend to depend on their parents for longer [100].

*Decreased sensitivity*: Species in which the majority of individuals are able to reproduce at any month of the year (year-round breeders); species that reproduce during the wet season but stop reproducing during the dry season. R-strategists would benefit because, in addition to a shorter pregnancy, they generally produce more offspring, which tend to grow at a faster rate to fully utilize the window of favorable environmental conditions with minimal (or no) parental care [100].

Sensory Detection of Fire Cues

Some animals are able to detect fire cues, either through olfactory (chemo-reception of smoke), visual (smoke plumes and flames), or acoustic (crackling sounds) means. Others may rely on thermoreceptors that can detect infrared radiation [31,101–104]. The greater the detection distance of fire cues, the greater the chance of an animal escaping and surviving. In general, olfactory cues can reach the farthest, followed by auditory and visual cues, which often signal immediate danger [7]. However, the value of fire cues as an early warning signal likely depends on an animal's sensory sensitivity, an individual's perceptual range, the fire behavior, and the environmental context [9,12].

*Increased sensitivity*: Species that spend most of their time surrounded by dense vegetation and rely primarily on the visual detection of fire may be particularly vulnerable, as the visual cues of fire might not enter an animal's perceptual range until it is very close [9,105,106]. Based on these terms, small-bodied animals could be even more vulnerable, as they usually have lower visual acuity [107].

*Decreased sensitivity:* Species that are able to detect olfactory and/or acoustic fire cues may have more time to make good escape decisions because they can detect fire at greater distances regardless of vegetation structure [37,39,101,108–110]. Species that are able to detect fire cues at lower thresholds (e.g., lower concentrations and from greater distances). Species that rely on thermoreceptors can detect infrared radiation from fires [102,111,112]. Species that spend most of their time on top of tall trees, in open or low-stature vegetation, and in topographically simple landscapes may have some escape advantage as the rising smoke plumes could enter the animal's perceptual range from a considerable distance (tens of kilometers), providing ample warning of fire risk [9,113].

Social Organization

Vigilance allows animals to monitor their surroundings and detect threats before it is too late to escape; however, it represents an allocation of time and energy that could be devoted to foraging and other fitness-enhancing activities [114]. In that case, sociality seems to be a good strategy because group members can reduce their investment in vigilance at no significant increased risk to themselves. As group size increases, individual vigilance decreases, and yet overall group vigilance and detection ability increases (effect of many eyes scanning for

threats) [115]. However, for collective detection to work, it is important that at least one individual is vigilant and detects a threat (in this case, fire), upon which it alerts other group members, whether seeking shelter, fleeing, or emitting alarm calls. The fear responses may be socially transmitted by a cascade effect or contagious alertness [116–118]. Therefore, individuals who have not detected the threat by themselves can use this information and still escape before it is too late [119,120]. Therefore, aspects of social organization, such as group size, relationship structure, and communication system can determine the effectiveness of collective detection of fire signals.

*Increased sensitivity*: Solitary species or those that live in small family groups (parents and young) rely on fewer individuals to detect approaching fire [121]. Species with poorly developed social relationships (e.g., groups with weak connections) and whose individuals or groups lack effective communication skills.

*Decreased sensitivity:* Gregarious species living in large groups [122–124]. Social species residing in more connected, reciprocal, and socially homogeneous groups [19,125–127].

### 3.1.2. Post Fire

Animals may be indirectly negatively affected after fire by the simplified habitat structure with greater microclimate extremes (e.g., temperature, humidity, and greater exposure to predators), diminished food resources (e.g., less food, lowered nutritional status, or decreased palatability), or interactions with other organisms (e.g., increased competition, predation, or parasitism) [128]. The physical and behavioral traits that can increase or decrease a species' sensitivity to the indirect effects of fire are described in detail below and illustrated in Figure 2.

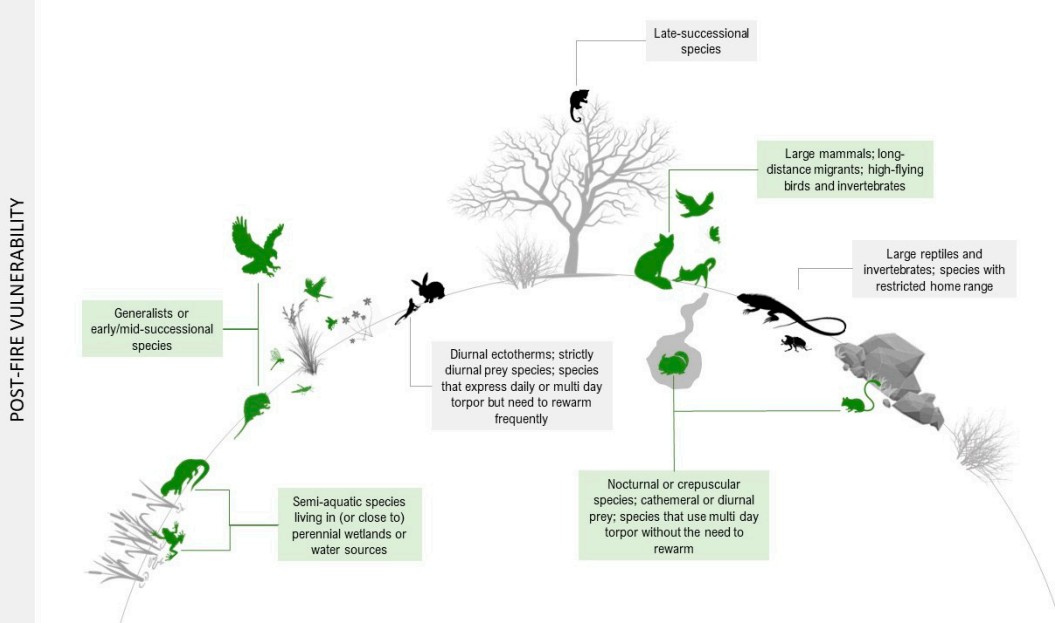

**Figure 2.** Fire vulnerability traits of species after a fire. In green are the animals most likely to survive in the post-fire landscape (decreased sensitivity). In black are the animals whose traits increase the probability of death after a fire (increased sensitivity).

Behavioral Plasticity

Population recovery depends on the species' behavioral plasticity with respect to habitat structure and diet.

*Increased sensitivity*: Late-successional species that require more structured habitats for nest sites and/or foraging, which take several years to recover [129]: canopy and upper-middle strata insectivores that forage in thicker bark or denser canopies [130,131]; small arboreal animals that depend on late successional resources (e.g., leaf litter and thick branches);

tree cavity-nesters that rely on highly-decayed wood or large living trees that provide both long-lasting cavities (e.g., in the main stem) and a series of single-use cavities (e.g., in dead branches) [131–133]; pollinators, nectarivores, and frugivores that benefit, respectively, from specific late-successional flowers, fruits and seeds of trees and shrubs [130,134–137]; low mesic insectivores that forage in thick litter [130]; saproxylic insects typically associated with large old trees and the decaying wood they generate [138]; invertebrates that have biological stages of their development inside fungal fruiting bodies [139].

*Decreased sensitivity*: Generalists that can temporarily adapt their diet and/or habitat preferences to the conditions and food resources available across the post-fire landscape [130]; species that may benefit from fire-induced changes such as predators (birds of prey) [140] and early or mid-successional species: open grassland species [141], aerial insectivores that benefit from the increased availability of flying insects [133,142,143]; nectarivores, frugivores, and granivores that forage on (or close to) the ground and benefit from the greater abundance of small herbaceous plants producing flowers, fruits, and seeds after fire [144–148]; deadwood-associated species [111].

Dormancy

Dormancy allows species to cope with the scorched post-fire environment, avoiding risky foraging movements within the simplified post-fire landscape and reducing the chances of starving or being captured by a predator [149–151].

*Increased sensitivity*: Species that express multi-day torpor but need to rewarm frequently. These species may deplete energy reserves and starve before their preferred habitat and resources recover since active rewarming from torpor requires a substantial increase in energy expenditure and can compromise energy savings gained from using torpor. On the other hand, passive rewarming from torpor involves basking in the sun and, consequently, being more exposed to predators [109,152]. Species that use daily torpor, which lasts only some hours rather than days or weeks, and is usually, but not always, interrupted by daily foraging and feeding. In this case, individuals will have to deal with the lack of food resources, which may impair the ability to rewarm after daily torpor [153]. Additionally, since torpid animals move slower than when normothermic and during foraging, they may be captured by a predator or exposed to altered environmental conditions [110].

*Decreased sensitivity*: Invertebrates that express aestivation and remain in an inactive stage that is remarkably resistant to water loss (e.g., mucus cocoon to resist desiccation) or that can afford the loss of water and sustain a dry form without compromising on revival upon rehydration (e.g., all anhydrobiotes) [154]; species that use multi-day torpor for weeks or even months after a fire or during fire season without the need to rewarm [38,150,151,153].

Endogenous Circadian Rhythms

Fire-induced changes may affect diurnal, crepuscular, and nocturnal species differently.

*Increased sensitivity*: Diurnal ectotherms that depend on thermoregulation opportunities [155,156]; strictly diurnal prey species, which become more vulnerable to increased predation rates [150,157–159].

*Decreased sensitivity*: For nocturnal or crepuscular animals, nighttime environmental temperatures are often lower than preferred temperatures. For this reason, individuals seek out warmer places within (or closer to) their preferred temperature range (e.g., deep bark fissures, hollow branches, warm rocks, trees, or inside retreat sites during the day), which are typically more protected from predators [160]. Cathemeral or diurnal prey that can adjust their daily activity patterns [150].

Mobility

Finding new habitat beyond the fire perimeter is likely to be a major determinant of population persistence because if individuals do not disperse they risk reduced fitness or increased mortality due to predation or starvation [150].

*Increased sensitivity*: Species with restricted home range (e.g., burrows or rock crevices); territorial species with high site fidelity that may perceive the risk of leaving their territory or home range to locate unburned patches to be greater than that of remaining in a familiar area with little or no food resource [41,161,162]. Migratory species (highly mobile) but with strong site fidelity to a limited number of stopover locations and travel routes can have adverse demographic results if traditional sites are completely scorched [163].

*Decreased sensitivity:* Highly mobile species that travel long distances (e.g., migratory or high-flying birds) or show metapopulation dynamics [104]; nomadic or non-territorial species with low site fidelity [161,162].

Morphology

*Increased sensitivity*: Large ectotherms. The body size–environment interaction is profound in ectotherms because they rely on external heat [164,165]. Since heat is dissipated more slowly in large-bodied animals (lower surface-to-volume ratio), being large in a post-fire environment may be particularly disadvantageous for an ectotherm as it can be more sensitive to overheating. Invertebrates with thinner cuticles are expected to desiccate faster [154].

*Decreased sensitivity:* Large mammals [166,167]; species with black morphs or that can change their color after a fire, likely diminishing predator detectability while foraging after a fire [168]; invertebrates with higher cuticle thickness, which gives the animal the advantage of reducing water loss (desiccation resistance) [169].

### 3.2. Fire Vulnerability Index

Fire exposure and sensitivity were combined to compute the Fire Vulnerability Index (FVI). Each species receives a score of "zero" or "one" per trait depending on whether or not it affects its sensitivity to fire. The sensitivity component (S) of the FVI is calculated by adding the values assigned to all traits, with a maximum value of 14. The exposure component (E) is calculated as the product of the values assigned to the three fire parameters listed in Table 1 (Equation (1)):

$$E_j = f_j \times e_j \times t_j \tag{1}$$

where *f* is the fire return interval, *e* is the fire extent, and *t* is fire seasonality in the area *j*.

The fire vulnerability index (FVI) for a given species *i* in each area *j* is a product of the two components (Equation (2))—the sensitivity of species *i* to direct and/or indirect effects of fire (S) and the exposure to fire across the area *j* (E):

$$FVI_{ij} = (S_i \times E_j / FVI_{max}) \times 10 \tag{2}$$

The maximum score for the FVI is 10 and those species with higher final scores will be positioned at the top of a continuous score ranking due to their greater sensitivity and exposure to fire. It is assumed that a sensitive species will not be threatened in an area where the fire regime is within a tolerable threshold. Alternatively, a resilient species may not decline even when experiencing some changes in fire regime patterns (Figure 3).

For management purposes, it may be necessary to define target species for monitoring and planning. In this case, we propose to classify species into four categories—"extremely vulnerable", "highly vulnerable", moderately vulnerable" and "least vulnerable"—according to the following thresholds: more than 7.5 (75%), 5.0 (50%), 2.5 (25%), and less than 2.5 (<25%), respectively. The thresholds correspond to possible scenarios of exposure and sensitivity. For example, the "extremely vulnerable" threshold is reached for species with high exposure and high sensitivity to fire. Figure 3 summarizes the scoring system.

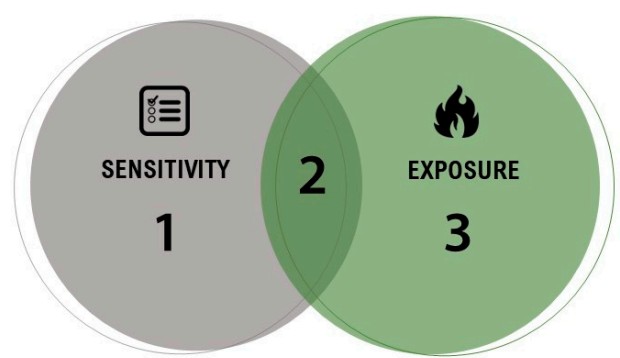

**1 - Sensitive, but not exposed -** *not currently at risk*
> Monitor fire exposure

**2 - Sensitive and exposed -** *at greatest risk*
> Monitor population trends
> Monitor fire exposure
> Specific research needed

**3 - Exposed, but not sensitive -** *may not be at risk*

## FIRE EXPOSURE

| FIRE RETURN INTERVAL | EXTENT | SEASONALITY |
|---|---|---|
| 0.5 Negligible | 0.5 Negligible | 0.5 Negligible |
| 1.0 Slight | 1.0 Slight | 1.0 Slight |
| 2.0 Moderate | 2.0 Moderate | 2.0 Moderate |
| 4.0 Important | 4.0 Important | 4.0 Important |
| 8.0 Extreme | 8.0 Extreme | 8.0 Extreme |

**FIRE EXPOSURE SUBSCORE ($E_i$)**

$$E_j = f_j * e_j * t_j$$

**OVERALL SCORE**

**FIRE VULNERABILITY INDEX**

$$FVI_{ij} = (S_i * E_j / FVI_{max}) * 10$$

## SPECIES'S SENSITIVITY TO FIRE

| DURING FIRE | POST FIRE |
|---|---|
| 0.0 Resistant/Resilient | 0.0 Resistant/Resilient |
| 1.0 Sensitive | 1.0 Sensitive |

**SPECIES'S SENSITIVITY SUBSCORE ($S_j$)**

$$S_i = \sum_{i=0}^{14}$$

▲ **7.50** / 10  **Extremely vulnerable**

▲ **5.00** / 10  **Highly vulnerable**

▲ **2.50** / 10  Moderately vulnerable

▼ **2.50** / 10  Least vulnerable

**Figure 3.** Scoring system for calculating the Fire Vulnerability Index.

## 4. Discussion

With the alarming pace of global climate change and the consequent multi-scale alterations in fire regimes, it is crucial to know which species are the most susceptible to local extinction due to the direct and indirect effects of fire. These species, which are potentially more sensitive to fire, could guide management decisions aimed at protecting biodiversity against the deleterious effects of wildfires. To address this issue, we have developed a fire vulnerability index that ranks animal species according to their sensitivity and exposure to fire.

We believe that assessments solely based on fire exposure or even on species sensitivity may be ineffective since species that are highly sensitive to fire should be considered more vulnerable when exposed to fire regimes with patterns very different from those for which they evolved or developed a suite of adaptations. Conversely, there are also species for which fire exposure is substantial, but their traits suggest that they may be able to cope with these better than other species. Thus, while monitoring and other conservation interventions might continue to be necessary, these species could represent a lower priority for fire-related conservation interventions in the immediate future.

A species' sensitivity, which is based on a species' traits, will change little over assessment timeframes, while exposure estimates, which depend on human actions and model predictions, will be more frequently revised. Because of this, fire vulnerability assessments can be updated based primarily on changes in exposure, making the index useful both as indices of change and for continually adapting management strategies.

Managers and researchers can take advantage of the continuous values associated with the ranking or classify species into four categories: "extremely vulnerable", "highly vulnerable", "moderately vulnerable", and "least vulnerable". It is recommended to focus attention primarily on extremely and, if possible, highly vulnerable species. However, case-by-case assessment of species' fire sensitivity and exposure also provides relevant information to tailor conservation interventions. For instance, species that have traits conferring high sensitivity to the direct effects of fires would be more likely to survive if management actions prioritize the maintenance of appropriate permanent refuges (e.g., forest patches or mature vegetation, wetlands, etc.) where individuals can shelter during a wildfire [17]. Fuel control can also be used to reduce fire intensity, minimizing damage to the tree layer [170]. Thus, nests in trees might escape smoke exposure and convective columns, while climbing species could take shelter in the upper strata of vegetation [25,59–62]. Lower-intensity fires are also expected to cause lower mortality to soil microbiota and animals sheltering in rock crevices, tree holes, peaty soil, and shallow burrows [43–46]. Fuel management through prescribed burns is recommended and widely used in fire-prone vegetation but should be considered with caution, as some species are still vulnerable even to low-intensity fire (e.g., leaf-litter invertebrates) [24]. In this sense, prescribed burns could be performed in smaller sections until the desired extension for the year is reached. For other species, even if individuals escape the fire, traits that confer high sensitivity to post-fire effects may increase susceptibility to predation or fire-caused changes in microclimates [128]. In such cases, securing temporary refuges (e.g., patches of unburned vegetation) within the fire perimeter could also be an effective management intervention, especially for smaller animals that rely on small-scale patches to meet their ecological requirements [9,15]. For species with specific habitat requirements (e.g., late-succession species), the protection and maintenance of mature, long-unburned vegetation in the landscape could be strongly recommended.

This study highlights the importance of broadening fire vulnerability assessment methods and introduces a new approach that considers biological traits that contribute significantly to a species' sensitivity alongside fire exposure. Since our index was initially designed for animal species inhabiting Brazilian savannas, it is desirable that it be properly adapted before being applied globally. Further refinement of our approach can provide several important contributions to the necessary management adaptations for any fire-prone ecosystem worldwide.

Further studies are needed to assess which and how functional traits and fire characteristics affect the ways in which animal species respond to fires. However, given the

difficulties associated with empirically validating all fire sensitivity traits and the urgency for conservation response to the growing threat of wildfires, the safest practical way is to apply the proposed fire vulnerability index to as many areas as possible as a starting point for monitoring and effectively implementing adaptive fire management.

**Author Contributions:** All authors contributed critically to the writing of the manuscript, developing technical content during and after the workshop and providing suggestions and comments in all versions. All authors have read and agreed to the published version of the manuscript.

**Funding:** This research was funded by the Brazilian Institute of Environment and Renewable Natural Resources (IBAMA) and the National Council for Scientific and Technological Development (CNPq) through the Ignite Project (441974/2018-0) and the EKLB postdoctoral fellowship (380006/2019-7). The workshop was partially funded by the General Office of University Extension (PROEX) of the Federal University of Minas Gerais (UFMG). The APC was funded by the University of São Paulo.

**Institutional Review Board Statement:** Not applicable.

**Informed Consent Statement:** Not applicable.

**Data Availability Statement:** Not applicable.

**Acknowledgments:** We thank the Pantanal Research Network, which was funded by the Ministry of Science and Technology (grant number: FINEP 01.20.0201.00) and whose team shared knowledge gained from the 2020 fires in the Pantanal with the intention of contributing to the debate on traits. Finally, we thank all reviewers for reading and providing valuable suggestions on earlier drafts of this paper.

**Conflicts of Interest:** The authors declare no conflict of interest.

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
