# Peer review of "In Case of Fire, Escape or Die: A Trait-Based Approach for Identifying Animal Species Threatened by Fire"

_fire, doi:10.3390/fire6060242_

Round 1
Reviewer 1 Report
I reviewed manuscript #2420722 “In Case of Fire, Escape or Die: A Trait-Based Approach to Identify Animal Species Threatened by Fire” submitted to the journal Fire by Eugenia Batista and 31 coauthors for a species issue on Recent Progress in Fire Ecology and Management in Tropical and Subtropical Ecosystems..
As the title of the manuscript indicates, it describes a trait-based framework utilizing components of exposure and sensitivity. The components are combined to form a Fire-Vulnerability Index for grouping animals into categories. The authors justify the need for the framework based on climate change affecting natural fire regimes and the influence of humans. Although the framework is oriented toward the Brazilian savannas but could be adapted to a range of large ecosystems with different fuels and fire regimes.
Overall, I found the manuscript very well organized and presented. The manuscript is well documented with references. Because the manuscript is basically a review of literature on information associated with fire effects on animals, there is little to comment on other than merits of rationale for the system developed and its application by resource managers. Particularly note worthy is the large number of coauthors (experts) involved in developing the framework.
My only suggestion for improvement of the manuscript would be consideration of providing an appendix demonstrating application of the framework for a small, selected group of savanna animals of contrasting vulnerabilities for exposure and sensitivity.
Author Response
Comment: My only suggestion for improvement of the manuscript would be consideration of providing an appendix demonstrating application of the framework for a small, selected group of savanna animals of contrasting vulnerabilities for exposure and sensitivity.
Response: Thank you for this suggestion. It would have been interesting to explore this aspect. However, in the case of our study, it seems slightly out of scope because here, we aim to propose a methodology for identifying locally vulnerable species. We gathered a substantial list of functional traits associated to the fire sensitivity in animal species. All this knowledge was previously scattered in the scientific literature, with low potential for application in the fire management and monitoring. For the first time, we connect the sensitivity of species to fire exposure through a numeric value (vulnerability index) capable of ranking species and identifying those with the greatest potential for local extinction in case of absence or inadequate fire management. Furthermore, adding an application section could compromise the publication of the article given the general limitations on the number of words.
Reviewer 2 Report
I enjoyed reading the manuscript, which is very informative and has great figures. The Fire Vulnerability Index is interesting and can be very useful to generate specific recommendations for the use of fire as a management tool in Brazilian savannas. In my opinion there are two main drawbacks in the paper that can be solved with better structure/explanation:
- The Methods are not totally clear and need some improvement (see specific comments below)
- The Methods focus on three variables: fire exposure, fire sensitivity and fire vulnerability index. However, the results are only about fire sensitivity. The Results should also include fire exposure and fire vulnerability index. Alternatively, the manuscript could deal only with fire sensitivity, because there is already a lot of information and the two of the figures.
The Introduction needs some revision of the text. For example, the sentence ‘However, extreme wildfires have resulted from a synergy between severe droughts, hot, low humidity and windy days, and human ignitions…’ could be improved to ‘severe droughts, high temperatures, low air humidity, frequent wind and increased human ignitions’.
Methods, 1st par. The sentence ‘which is inherently mediated by resistance and adaptive capacity or resilience’ is unclear. Comprehension of exposure and sensitivity would benefit from more concrete ideas and examples of traits using some well-known animal species.
2.1. Line 3 ‘burning the ground layer’. Is it the ground that is burned? Or is it the litter and the grass layer? After this, the text can be improved to ‘burning the shrubs and lower tree branches’ (‘leaves’ is confusing here)
2.1. Lines 4-5 ‘naturally extinguished at forest edges, where there is low fuel’ what do you mean? This is contrary to the concept of forest. Do you mean that vegetation structure of these forests is unfavourable to fire spread?
2.1. 3rd par. The expressions ‘We propose to evaluate this parameter…’ ‘We recommend a 20-year period for analysis’ seem out of context in a Methods’ section. If this is what you have done, the sentences should be ‘We evaluated…’, ‘We analysed a 20-year period’, etc.
2.1. 3rd par. The names of certain variables do not correspond to their definitions and units. Fire return interval is a variable that is measured in years. The text define it as the “the percentage of fire-prone vegetation that is outside the expected thresholds for natural fire regimes”. This is not the concept of ‘interval’. Also, Fire extent is a spatial variable, but you define it as “the number of times a study area burned… over a period of 10 years”. This makes the methods a bit confusing. Names should be changed to match the variable definitions.
Table 2. How have the scores been attributed? With current scores an extreme effect on vulnerability is only 1.7 times larger than a moderate effect. This is not what one can expect when comparing a moderate to an extreme effect. I would propose a geometric scale: 0.5-1-2-4-8 or a similar one. The FVI should be adapted consequently.
2.2. This section is confusing to me. First of all, concrete details about the number of participants on workshops and consultations should be given, together with the keywords used for the literature search. Second, it follows a summary of traits that are then developed under the result section. In my opinion this is not a good structure because you providing information in Methods that you then consider to be results. You could either delete all this part about traits or just present it as a short background (with a few citations) to the workshops and search. In any case, the text should be revised because it seems more suited for the Introduction than for a Methods section.
Figure 1 is great and helps a lot to understand the method of computing FVI.
Section 3.1. I would add the vulnerability to predation by opportunistic predators while escaping the fire front.
Section 3.2.1 I would add the capacity to use novel foods that can be available shortly after fire (plant regrowth, flowers and seeds, some invertebrate species…) but are not found in unburned areas.
Section 3.2.5 I would add fire melanism, i.e. species with black morphs that are more frequent shortly after fire because they have lower mortality from predation. See, for example: Karlsson, M., Caesar, S., Ahnesjo, J., & Forsman, A. (2008). Dynamics of colour polymorphism in a changing environment: fire melanism and then what? Oecologia, 154, 715-724.
The English needs only minor revision, although I am not a native English speaker. See my suggestions to improve some sentences in the former comments.
Author Response
Comment 1: The Methods focus on three variables: fire exposure, fire sensitivity and fire vulnerability index. However, the results are only about fire sensitivity. The Results should also include fire exposure and fire vulnerability index. Alternatively, the manuscript could deal only with fire sensitivity, because there is already a lot of information and the two of the figures.
Response: Thank you for pointing this out. We partially agree with this comment. To improve the structure, we moved the "Fire Vulnerability Index" to the results section, since it is the main purpose of the study. However, we believe that fire exposure should not be in the results section because it should apply locally, varying according to each region.
Comment 2: The Introduction needs some revision of the text. For example, the sentence ‘However, extreme wildfires have resulted from a synergy between severe droughts, hot, low humidity and windy days, and human ignitions…’ could be improved to ‘severe droughts, high temperatures, low air humidity, frequent wind and increased human ignitions’.
Response: It was addressed.
Comment 3: 2.1. Line 3 ‘burning the ground layer’. Is it the ground that is burned? Or is it the litter and the grass layer? After this, the text can be improved to ‘burning the shrubs and lower tree branches’ (‘leaves’ is confusing here)
Response: It was addressed.
Comment 4: 2.1. Lines 4-5 ‘naturally extinguished at forest edges, where there is low fuel’ what do you mean? This is contrary to the concept of forest. Do you mean that vegetation structure of these forests is unfavourable to fire spread?
Response: It was addressed.
Comment 5: 2.1. 3rd par. The expressions ‘We propose to evaluate this parameter…’ ‘We recommend a 20-year period for analysis’ seem out of context in a Methods’ section. If this is what you have done, the sentences should be ‘We evaluated…’, ‘We analysed a 20-year period’, etc.
Response: You have raised an important point here. However, our proposal in this study is not to apply the index.
Comment 6: 2.1. 3rd par. The names of certain variables do not correspond to their definitions and units. Fire return interval is a variable that is measured in years. The text define it as the “the percentage of fire-prone vegetation that is outside the expected thresholds for natural fire regimes”. This is not the concept of ‘interval’. Also, Fire extent is a spatial variable, but you define it as “the number of times a study area burned… over a period of 10 years”. This makes the methods a bit confusing. Names should be changed to match the variable definitions.
Response: We agree with this and have incorporated your suggestion throughout the manuscript.
Comment 7: Table 2. How have the scores been attributed? With current scores an extreme effect on vulnerability is only 1.7 times larger than a moderate effect. This is not what one can expect when comparing a moderate to an extreme effect. I would propose a geometric scale: 0.5-1-2-4-8 or a similar one. The FVI should be adapted consequently.
Response: Agree. We have, accordingly, changed the fire exposure scores to emphasize this point.
Comment 8. 2.2. This section is confusing to me. First of all, concrete details about the number of participants on workshops and consultations should be given, together with the keywords used for the literature search. Second, it follows a summary of traits that are then developed under the result section. In my opinion this is not a good structure because you providing information in Methods that you then consider to be results. You could either delete all this part about traits or just present it as a short background (with a few citations) to the workshops and search. In any case, the text should be revised because it seems more suited for the Introduction than for a Methods section.
Response: It was addressed.
Comment 9: Section 3.1. I would add the vulnerability to predation by opportunistic predators while escaping the fire front. Section 3.2.1 I would add the capacity to use novel foods that can be available shortly after fire (plant regrowth, flowers and seeds, some invertebrate species…) but are not found in unburned areas. Section 3.2.5 I would add fire melanism, i.e. species with black morphs that are more frequent shortly after fire because they have lower mortality from predation. See, for example: Karlsson, M., Caesar, S., Ahnesjo, J., & Forsman, A. (2008). Dynamics of colour polymorphism in a changing environment: fire melanism and then what? Oecologia, 154, 715-724.
Response: It was addressed.
Reviewer 3 Report
This is a review the effects of fire on animals and introduction of a proposed fire vulnerability index to identify animal species that are most at risk from fire. The authors have compiled a commendable review of recent literature on fire effects on animals. I have the following general suggestions and comments.
The Fire Vulnerability Index (FV\I) usefulness to managers and researchers as "indices of change" to continually adapt management strategies would benefit from the addition of a case study or examples that would help readers better understand how to make the proposed conceptual model operational. Since the index "was initially designed for animal species inhabiting Brazilian savannas," it would be immensely helpful to calculate FVI values for a suite of species from those savannas, apply it to a realistic management situation, and interpret the results.
Another more general concern is combination of sensitivity (15 animal functional traits or life history characterisitcs) and exposure (fire return interval, extent, and seasonality) into the single index value. These variables could behave in unexpected ways that get lost in the one value. It might be useful to explore interactions among these variables using multivariate statistics and graphs.
Page 3. Introduction. Suggested revision: Natural fire has shaped species evolution in savanna ecosystems worldwide. In these ecosystems, animal species are relatively tolerant of low-severity and patchy fires.
Figures. The labels on the left-hand side of the figures gives the impression that these are graphs. The arc of the figures starting in wetlands on the left suggests topography. Maybe just pair opposites in two columns: Increased sensitivity and Decreased sensitivity. Pictorial representations of the traits (e.g., strong fliers vs. weak fliers) would be side-by-side. Just a thought.
Author Response
Comment 1: Since the index "was initially designed for animal species inhabiting Brazilian savannas," it would be immensely helpful to calculate FVI values for a suite of species from those savannas, apply it to a realistic management situation, and interpret the results.
Response: Thank you for this suggestion. It would have been interesting to explore this aspect. However, in the case of our study, it seems slightly out of scope because here, we aim to propose a methodology for identifying locally vulnerable species. We gathered a substantial list of functional traits associated to the fire sensitivity in animal species. All this knowledge was previously scattered in the scientific literature, with low potential for application in the fire management and monitoring. For the first time, we connect the sensitivity of species to fire exposure through a numeric value (vulnerability index) capable of ranking species and identifying those with the greatest potential for local extinction in case of absence or inadequate fire management. Furthermore, adding an application section could compromise the publication of the article given the general limitations on the number of words.
Comment 2: Another more general concern is combination of sensitivity (15 animal functional traits or life history characterisitcs) and exposure (fire return interval, extent, and seasonality) into the single index value. These variables could behave in unexpected ways that get lost in the one value. It might be useful to explore interactions among these variables using multivariate statistics and graphs.
Response: You have raised an important point here. However, we believe that, a numerical index could already represent the relative vulnerability of the species.
Comment 3: Page 3. Introduction. Suggested revision: Natural fire has shaped species evolution in savanna ecosystems worldwide. In these ecosystems, animal species are relatively tolerant of low-severity and patchy fires.
Response: It was addressed.
Comment 4: Figures. The labels on the left-hand side of the figures gives the impression that these are graphs. The arc of the figures starting in wetlands on the left suggests topography. Maybe just pair opposites in two columns: Increased sensitivity and Decreased sensitivity. Pictorial representations of the traits (e.g., strong fliers vs. weak fliers) would be side-by-side. Just a thought.
Response: Thank you for this suggestion, but the purpose of the figure is just to illustrate the vulnerability traits.
Round 2
Reviewer 3 Report
I fail to see how fitting real-world knowledge of species to the proposed index and evaluating its usefulness to a management situation is "out of scope." Proposing a complex index without going through the process of applying it seems incomplete. Problems will arise with any new methodological approach. Shouldn't the original authors be the ones to identify challenges with their index and explore how best to work around them?
Given the problem of the allowable length of the manuscript for this journal, perhaps a way to address my concerns is to create two papers: the index and an application. Part 1 and Part 2.
Author Response
For this first paper, we focused on the literature review and index proposition. We are already working on a second applied paper, including the development of a tool to make it easier for other researchers to apply the methodology.